# Snowy Dove: An open-sourcetoolkit for pre-processing of Chinese Gaofen series data

**Debao Yuan[1], Liuya Zhang[1]\*, Jiantao Dong[2], Cheng Fan[3], Xurui Zhao[1]**

**1** College of Geoscience and Surveying Engineering, China University of Mining and Technology (Beijing), Beijing, China, **2** Satellite Application Center for Ecology and Environment, Ministry of Ecology and Environment of People's Republic of China, Beijing, China, **3** State Environmental Protection Key Laboratory of Satellite Remote Sensing & Key Laboratory of Remote Sensing and Digital Earth, Aerospace Information Research Institute, Chinese Academy of Sciences, Beijing, China

\* zhangliuya0728@126.com

## Abstract

Since the launch of the Chinese High-resolution Earth Observation System (CHEOS) program, China has strengthened its research and development in the field of satellite remote sensing. A large number of sensors has been or will be launched, providing very large data streams which all require processing of the engineering data, as provided by the instruments, to physical data which will be used for further processing and interpretation. To handle such large data streams we developed a one-click batch pre-processing toolkit for CHEOS remote sensing data as described in this paper. In this toolkit, IDL language and environment are used as the primary program combined with other programming languages developed in this research. In this paper, we first describe the Gaofen (GF) series data used in this research and then introduce the function design and realization of this one-click batch pre-processing toolkit. Some examples will be presented to illustrate the application of the toolkit to data from several CHEOS satellites.

## Introduction

The Chinese High-resolution Earth Observation System (CHEOS) program was approved by the Standing Committee of the State Council and launched in 2010. The program aims to accelerate the development of China's spatial information and application technology. This program will build a high-resolution advanced earth observation system based on satellites, stratospheric airships and aircraft, improved ground resources, and join other means of observation to form a stable operation system with space-time coordination, all-weather, all-day and global observation capabilities and meet the needs of national economic construction, social development and national security [1–3]. An important part of the CHEOS program is the Gaofen series of Chinese civilian remote sensing satellites. Gaofen data set are used in this study.

CHEOS satellite data have been widely used for many applications [4]. All applications require pre-processing of the L1A data, which are the unprocessed raw data with time reference, ancillary information (including radiation, geometric correction factors, etc). The

**Data Availability Statement:** Some third-party data were used in the research of this manuscript, and the completeness and accuracy of these data are essential to support the conclusions of this study. In accordance with PLOS ONE's data policy,

**Funding:** This work was supported by Hebei Natural Science Foundation Ecological Smart Mine Joint Fund (No. E2020402086), and this work was also supported by National Natural Science Foundation of China(NSFC)(No.52174160). The funders had no role in study design, data collection and analysis, decision to publish, or preparation of the manuscript.

**Competing interests:** NO authors have competing interests Enter: The authors have declared that no competing interests exist.

preparation of L1A data for further processing to L2 data, which are the basis for further application of satellite data in scientific and applied studies, is referred to as pre-processing which is the subject of this paper. Pre-processing includes geometric precision correction, radiation correction, image mosaic and cropping, atmospheric correction, etc. It is a necessary step for the application of satellite remote sensing data [5]. Generally, the pre-processing of remote sensing data is carried out image by image, using ENVI software which is operated step by step. This is not only time-consuming, but also generates a large amount of redundant data during processing, which is not conducive to efficient and convenient processing of images. If data pre-processing can be performed efficiently in batches, then remote sensing technology can provide flexible and cost-effective solutions for practical applications such as monitoring of environmental pollution.

Some pre-processing studies have been made for the Chinese satellites, as shown in the following examples. The first example is a remote sensing monitoring data processing system which has been developed by Qin et al [6] to efficiently detect the river construction projects in Guangdong, the system has complete functions and realized the pre-processing and subsequent detection and mapping functions. Another example is the development of software for measuring the grape growing area, which is an application-oriented project by Song et al. [7] who used used C# and Arc Engine for this purpose. A third examples is the batch pre-processing of domestic high-resolution satellite images based on IDL developed by Wang et al. [8], but only for GF-1 and GF-2 and the codes are not open-source. Other examples are the use of Fortran and IDL to complete the GF-1 atmospheric correction batch program using the 6S radiative transfer mode, which can be stably used for agricultural remote sensing monitoring business processes Liu et al. [9], and the batch processing tool for domestic satellite images designed by Li et al. [10], but only for ZY02C, ZY3 and GF-1 satellite data. At present, there is no tool that can realize the batch processing of all the pretreatment processes of GF1/2/6, etc. The paper presents a novel toolkit that integrates and optimizes Gaofen data preprocessing steps, significantly reducing processing time and enabling batch processing for GF1/2/6 data, which is a unique advancement over existing tools like ENVI.

In this paper, we propose a new toolkit, named Snowy Dove, for pre-processing of GF series satellites, including GF-1B/C/D, GF-2 and GF-6 (see Table 1). The datasets will be introduced in Section 2. The function and the flowchart of the Snowy Dove are described in Section 3. The implementation of the software and examples of the application are presented in Section 4. Snowy Dove is discussed in Section 5 and conclusion are presented in Sections 6.

## Materials and methods

### Date

The GaoFen imagery used in this research is supported by the China Centre for Resources Satellite Data and Application (http://www.cresda.com, accessed on 5 July 2024). Information on

**Table 1. The data supported in Snowy Dove are cited.**

| Platform | Launch Time | Sensor | Band Setup | Resolution(m) | Quantization bits |
|---|---|---|---|---|---|
| GF-1 | 2013 | PMS1-2 | PAN+4MSS | PAN 2, MSS 8 | 10 |
| | | WFV1-4 | 4MSS | 16 | |
| GF-2 | 2014 | PMS1-2 | PAN+4MSS | PAN 0.8, MSS 3.2 | 10 |
| GF-1B/C/D | 2018 | PMS | PAN+4MSS | PAN 2, MSS 8 | 12 |
| GF-6 | 2018 | PMS | PAN+4MSS | PAN 2, MSS 8 | 12 |
| | | WFV | 8MSS | 16 | |

the platforms and sensors supported in Snowy Dove are presented in Table 1, together with the band setup (where PAN and MSS represent panchromatic bands and multi-spectral band, respectively) and the pixel resolution.

### GF-1

Gaofen-1 (GF-1) is the first satellite in the CHEOS program [11], which was successfully launched on 26 April 2013. It flies in a sun synchronous orbit at an altitude of 645 km, and its local overpass time in the descending orbit is 10:30 am. It has two Panchromatic & Multispectral Scanners (PMS) named PMS1 and PMS2, and four Wide Field of View Multispectral Cameras (WFV) named WFV1, WFV2, WFV3 and WFV4. The PMS sensors collect data in 5 spectral bands, including a panchromatic band with a spatial resolution of 2 m and 4 multi-spectral bands (represent blue, green, red and near-infrared, respectively) with a spatial resolution of 8 m. The WFV sensor collects data in 4 multi-spectral bands with a lower spatial resolution of 16 m, but with relatively wider swath. The number of quantization bits of both the PMS and the WFV sensor is 10, i.e., the integer dynamic range of the imageries is from 0 to 1023 (210–1). The only difference between the PMS and WFV sensors is that the center wavelengths are different.

Gaofen-1B, Gaofen-1C and Gaofen-1D (GF1-B/C/D), was successfully launched on 31 March 2018, on one bus. A PMS sensor is onboard each of these three satellites. The spatial resolution and band setup of this PMS sensor is identical to the one onboard GF-1, but with a higher quantization bit of 12, that means the integer dynamic range of the imageries is from 0 to 4095 (212–1), while the PMS sensor onboard GF-1 is from 0 to 1023.

### GF-2

Gaofen-2 (GF-2) was successfully launched on 19 August 2014. It has two Panchromatic & Multispectral Scanners (PMS) named PMS1 and PMS2 [12]. The band setup and quantization bits of the PMS sensor onboard GF-2 is identical to the one onboard GF-1, but the spatial resolution of the panchromatic band is enhanced to 0.8 m and for the multi-spectral bands it is enhanced to 3.2 m. The GF-2 satellite is similar to GF-1, with the only difference that the center wavelengths between the two PMS sensors are different.

### GF-6

The Gaofen-6 (GF-6) satellite was launched successfully on 2 June 2018 [13]. GF-6 is an optical remote sensing satellite with red-edge band. It is in a 645 km sun synchronous orbit, and its local overpass time in the descending orbit is 10:30 am. It has one PMS sensor and one WFV sensor [14]. The band setup, spatial resolution and quantization bits of the PMS sensor onboard GF-6 is identical to the PMS sensor onboard GF-1B/C/D. The WFV sensor onboard GF-6 has 8 bands (representing wavelengths in the blue, green, red, near-infrared, red-edge I, red-edge II, coastal and yellow) with a spatial resolution of 16 m and 12 quantization bits.

## Pre-processing function design and results

The Snowy Dove pre-processing toolkit consists of two parts: a preparation part and a processing part. These two parts are shown as separate blocks in the flow chart of Snowy Dove presented in Fig 1.

The preparation part of Snowy Dove checks the validity of the input parameters, the operating environment and it takes care of the initialization to ensure the correct environment necessary for the processing part. In the processing part the data handling takes place. Such as

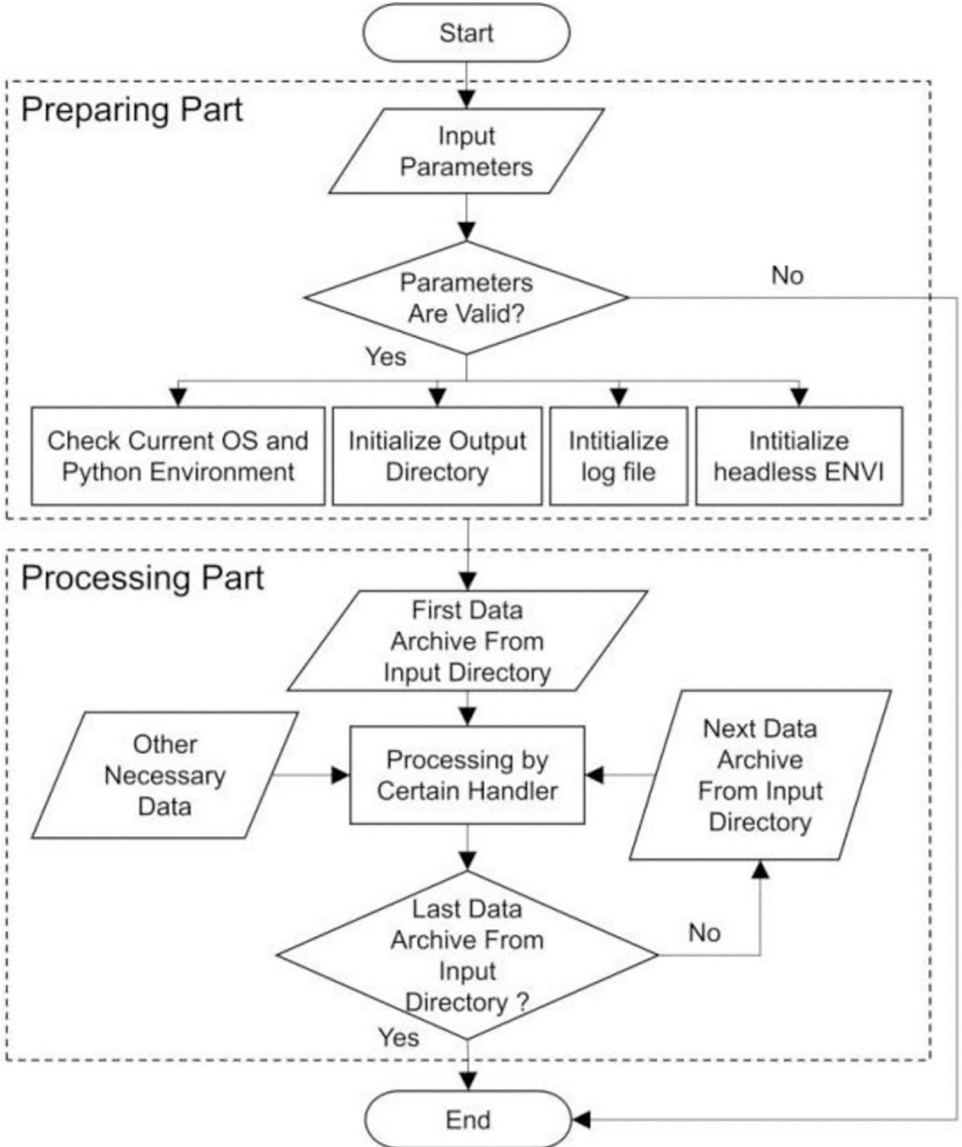

**Fig 1. Snowy Dove flow chart.**

calculation and correction. For each type of sensor, the pre-processing components are slightly different. The flow chart for the pre-processing of data obtained by the PMS sensors onboard GF-1, GF-2 and GF-6 is shown in Fig 2. Each of the steps shown in the flow chart is explained in the following sections.

## Data decompression

The GF series data from the data distribution agency are usually named as "<satellite name->_<sensor name>_<longitude>_<latitude>_<date>_<product number>.tar.gz",eg. GF1C_PMS_E119.5_N29.1_20190126_L1A1021362773.tar.gz. It is necessary for the user to first decompress the data file (filename.tar.gz to filename.tar) and then decompress the tar file to the original data. The original data includes the L1A level tiff image and its corresponding Rational Polynomial Coefficients (RPC) file.

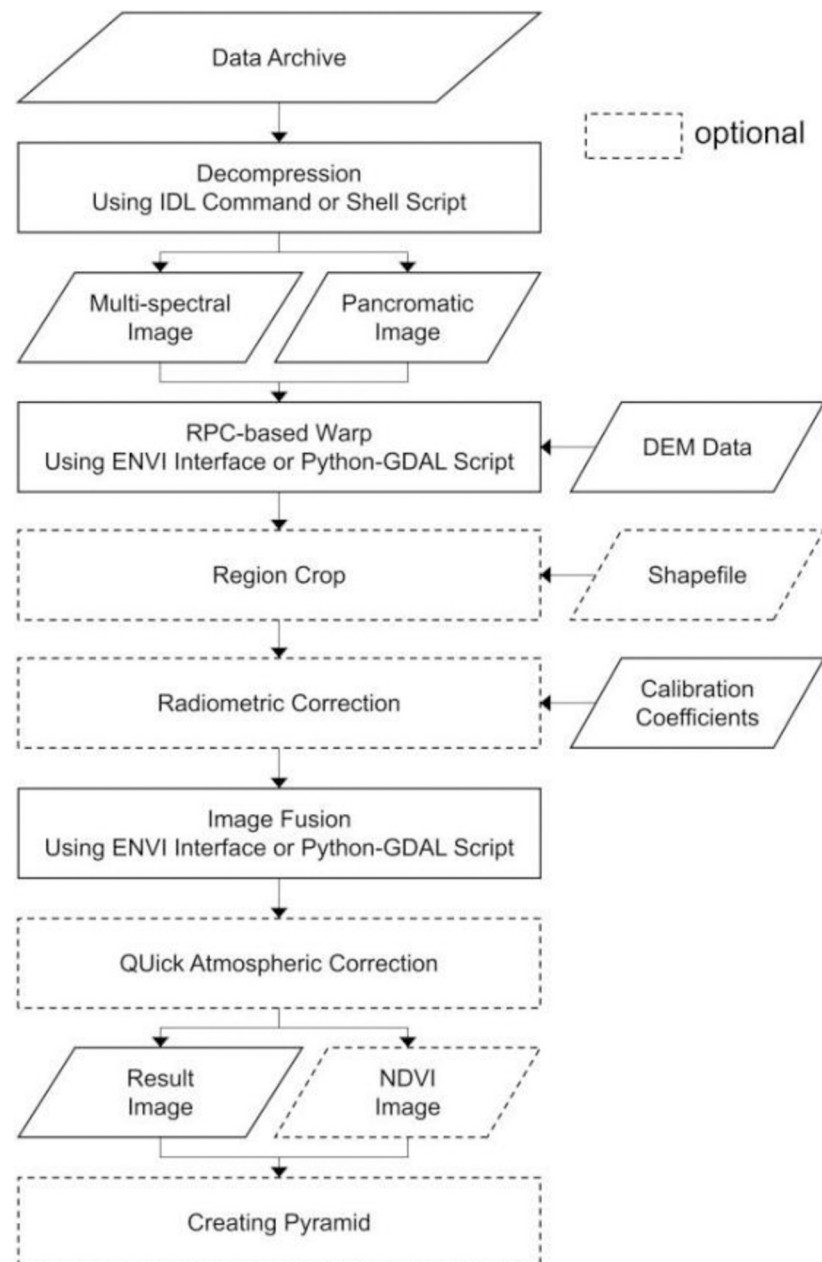

**Fig 2. Flow chart for pre-processing of data obtained with the PMS sensors onboard GF-1, GF-2 and GF-6.**

## Orthorectification

Changes in the terrain of the earth's surface and the tilt angle of the satellite or aerial sensor will affect the size of the elements displayed on the satellite image. The more diverse the terrain, the more serious the inherent distortion of the image.

Before collecting information from remote sensing images useful for mapping or geographic information systems (for example, road, vegetation and water features), it is necessary to eliminate distortion problems geometrically and orthorectify the image, i.e. create a flat image at each location, with consistent proportions on all parts of the image. Without this

process, users will not be able to directly and accurately measure distance, angle, location and area.

The RPC files provides a compact representation of a ground-to-image geometry, which allows for photogrammetric processing without requiring a physical camera model. The information in the RPC files is used together with Digital Elevation Model (DEM) to generate accurate orthorectified raster data. A DEM is a 3D computer graphics representation describing the variation of elevation across the terrain. DEM are often used in geographic information systems, and are the most common basis for digitally produced relief maps.

### Region crop

Remote sensing images are usually based on scenes. When the area of interest is part of a scene, the data needs to be cropped to retain only the area of interest. The orthophoto remote sensing image is cropped based on the geographic location through the shape file, to remove the uninteresting area and thus improve the efficiency of image processing.

### Mosaicking

Both GF-1B/C/D PMS data and GF-6 WFV data are tiled data, that is, the compressed file contains more than one image, and these images are located adjacent to each other to form a scene image. If the research area spans more than one of the images, the images that cover the research area need to be stitched together to combine the separate images into a single image.

### Radiometric correction

For L1A GF data, the values stored in the image are electrical signals formed by discretization of optical signals. For professional researchers, the original dimensionless Digital Number (DN) value needs to be converted into a numerical value with physical units through a certain algebraic conversion. This process is called radiometric calibration.

The calibration usually takes place pre-launch. After launch, the sensor degrades with time, which may be in an optical or electrical sense or in the supportive components. The degradation may occur during launch, due to orbital jumping, due to cosmic ray bombardment etc. Therefore, calibration needs to be checked regularly. For the GF series data, the China Resources Satellite Application Center provides an updated version of linearized calibration coefficients every year to ensure the normal application of satellite data.

$$Le = Gain * DN + Bias \qquad (1)$$

Le is the equivalent radiance at the entrance pupil of the satellite load channel (in W·m-2·sr-1·μm-1). Gain and Bias are the calibration coefficient gain and offset, respectively, in the same units as Le.

### Image fusion

An image observed by a PMS sensor includes multi-spectral scanning (MSS) data and panchromatic (PAN) data. MSS only observes the earth in the red, green, blue and near infrared channels. However, PAN integrates over all MSS channels and therefore receives more energy. Since the amount of solar radiation collected by each PAN pixel is higher than that collected by MSS pixels, PAN can detect brightness changes in a smaller spatial range, i.e. PAN pixels can be smaller than MSS pixels.

To take advantage of the better spectral resolution of MMS and the better spatial resolution of PAN for better use in drawing and analysis, the two are often combined or merged to

improve the visual effect of the image. The main idea of the image fusion algorithm is to retain the high-frequency signal components of PAN and replace the low-frequency signal components with those of MSS to generate high spatial resolution multi-spectral images.

## Atmosphere correction

Atmospheric correction was then applied to the calibrated satellite images to reduce the effect of atmospheric absorption and scattering. It is very important for land surface remote sensing and application. Atmosphere correction is the elimination of atmospheric effects on the reflectance at the top of the atmosphere which, in land surface remote sensing, is used to calculate the surface reflectance from remote sensing images. Surface reflectance retrieval is an important step in the data processing chain for the extraction of quantitative information in many applications [15].

Atmosphere correction algorithms usually include two main steps. First, the optical characteristics of the atmosphere are estimated by using special features of the earth's surface, or by directly measuring atmospheric composition, or by using physical models. Next, quantities related to atmospheric correction are calculated using a radiative transfer algorithm. Finally, the remote sensing image is corrected by the inversion program that calculates the surface reflectance.

The reflectance usually ranges from 0 to 1, which is floating point data. However, in most atmospheric correction software, in order to reduce the storage space of the image, the reflectance value is expanded by 10000 times by default to represent the surface reflectance as an integer. For example, when, after atmospheric correction, the surface reflectance of a pixel is 0.0234, after enlargement the surface reflectance of the corresponding pixel is represented as an integer with a value of 234. For practical use, this value needs to be divided by 10000 to express the true surface reflectance.

In this study, Atmospheric correction was applied using the Quick Atmospheric Correction (QUAC) algorithm [16–18]. QUAC is an atmospheric correction that requires only approximate specification of sensor band locations. It uses a scene approach, and thus, it is faster than are corrections with first principle radiative models. The QUAC principle assumes that the average spectral curve of several (>10, typically 50) diverse materials from an image should have the same spectral signature as precalculated 'universal' signature derived by averaging diverse collection of reflectance spectra from the spectral library. If there is a difference between the average library spectrum and the average from the observed endmembers spectrum, it represents an effect of the atmosphere.

## NDVI calculation

The Normalized Difference Vegetation Index (NDVI) is an indicator of the amount of green vegetation in a pixel. It is calculated from the reflectances measured in a near-infrared (NIR) and a red (R) channel:

$$NDVI = (NIR - R)/(NIR + R) \tag{2}$$

The NDVI ranges from -1 to 1. If the NDVI is close to 1, it is likely to be dense vegetation, when the NDVI is close to 0, there is no vegetation.

## Format conversion

The Interactive Data Language (IDL) is widely used in remote sensing data processing. It has the characteristics of cross-platform, i.e. it does not depend on the operating system and

hardware environment, and can perform fast array operations, which is very useful for processing of satellite images. In the ENVI/IDL environment, the pre-processed image includes a file that only stores pixel values and a header file that stores image information. Although this image format can be opened and operated in most GIS software or remote sensing software, it is not universal. For example, if the researcher wants to manipulate the pre-processed data in Adobe Photoshop (PS), the PS cannot recognize this ENVI format image. To facilitate this and thus improve the generality of the results, the pre-processed results can be converted to Geo TIFF or BigGeo TIFF.

## Implementation of Snowy Dove

Snowy Dove can be implemented in two ways. The first way is in the IDL Development Environment (IDLDE): complete pre-processing by calling binary sav files. The second way is by using the IDL interpreter in the terminal (*nix environment) or Power Shell (windows environment) to call the script and pass in the parameters to complete the pre-processing. The implementation of one-click pre-processing in IDLDE is described in the S1 Appendix.

Snowy Dove has been systematically tested using data from the supported GF series data. The results for four different satellites, GF-1 WFV, GF-6 WFV, GF-1 PMS, GF-2 PMS, are presented in Figs 3–6. Each figure consists of three parts. Figures (a) show the original image data, figures (b) show the result of orthorectification and radiative correction and figures (c) show the NDVI corresponding to figure (b). Fig 3 shows results from an GF-1 WFV3 image, centered around location 94.7˚E, 27.3˚N, on 8th Jan 2019. Fig 3A shows the original image, which is tiled after orthorectification (Fig 3B) due to the rotation of the earth. The orthorectified image has a higher contrast than the original image and therefore shows more detail. In the NDVI image, the water body has a lower value, so it appears darker, which will help to classify the surface features. Fig 4 shows the result from a GF-6 WFV image, centered around location 115.9˚E, 38.0˚N, on 7th Apr 2019. Fig 4A is the original image, which is composed of three parts, so the image mosaic is required first. The images after stitching have good consistency in spatial distribution. Moreover, in the NDVI image, the contrast between vegetation and water body area is obvious. Figs 5 and 6 show images collected by the PMS sensors mounted on the GF-1 and GF-2, respectively. Fig 5 is the result from a GF-1 PMS2 image, centered around location 95.0˚E, 29.6˚N, on 22nd Nov 2018. Fig 6 is the result from a GF-2 PMS2 image,

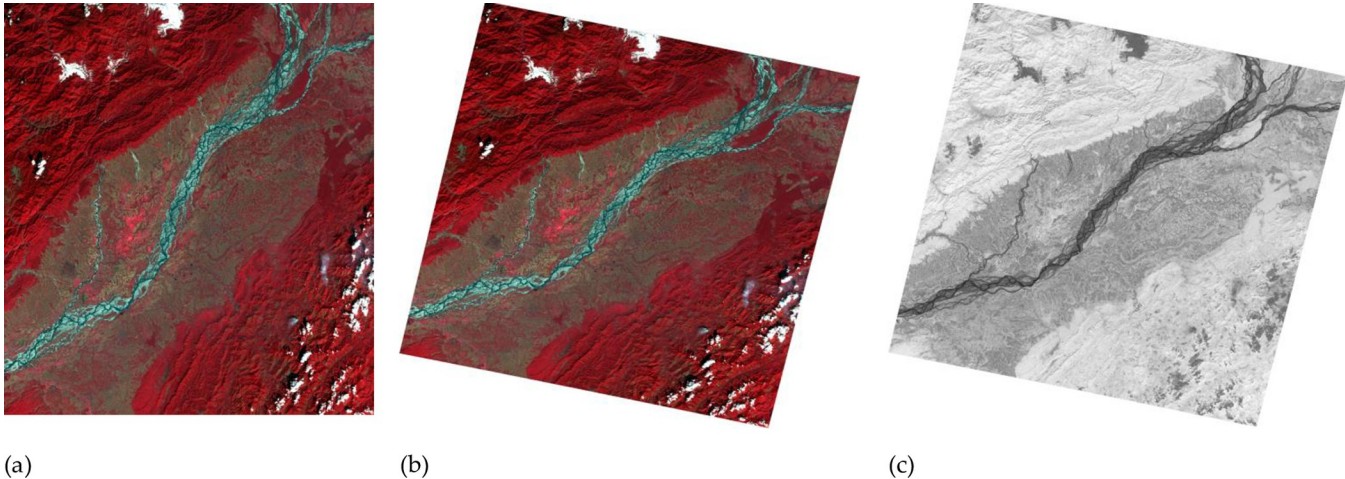

(a)                                    (b)                                    (c)

**Fig 3.** GF-1 WFV3 images with center location 94.7˚E, 27.3˚N, recorded on 8th Jan 2019: (a) Original image; (b) orthorectified image; (c) NDVI.

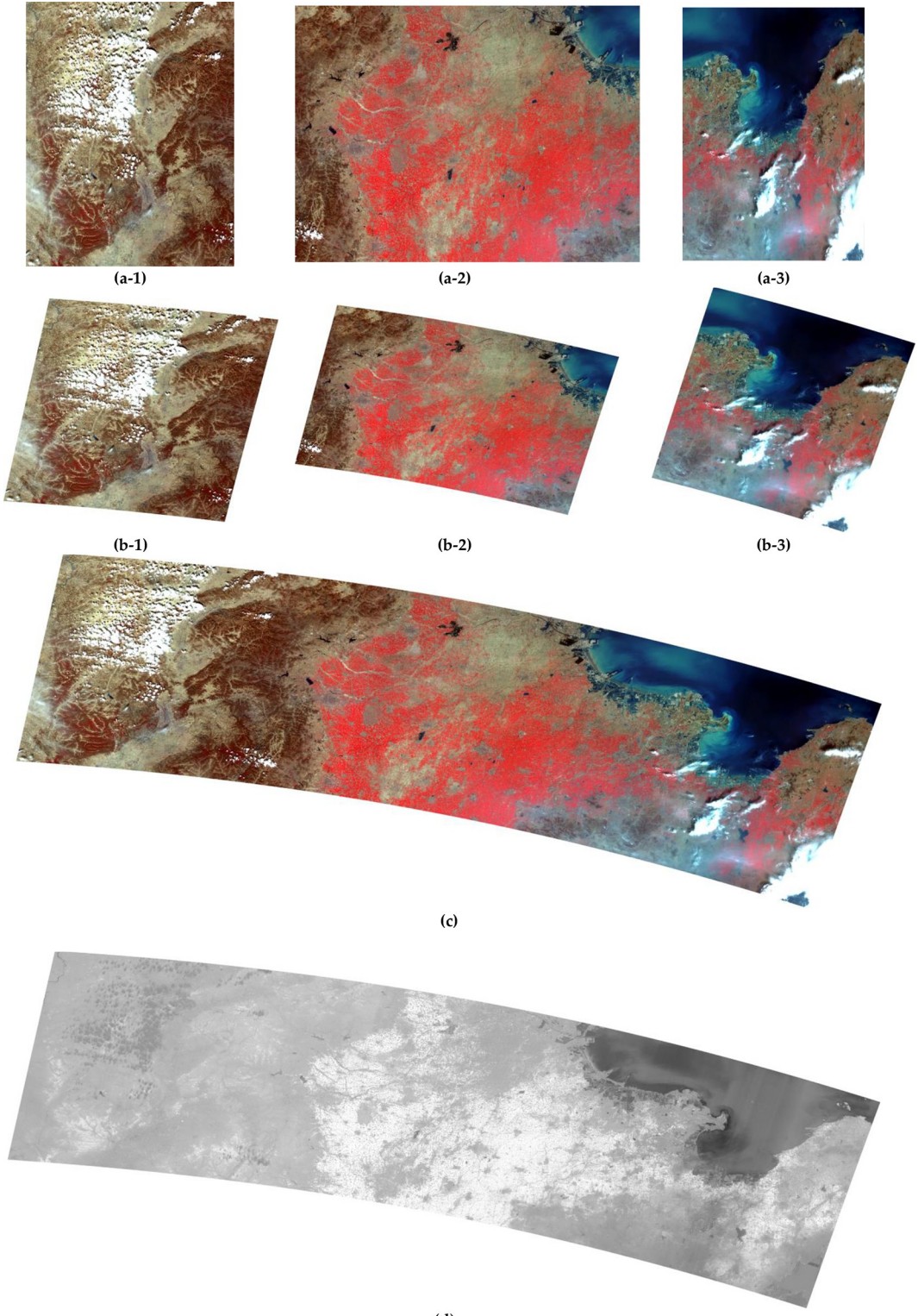

**Fig 4.** GF-6 WFV image with center location 115.9°E, 38.0°N, recorded on 7th Apr 2019: (a-1/2/3) Original images; (b-1/2/3) orthorectified image separately; (c) orthorectified image; (d) NDVI image.

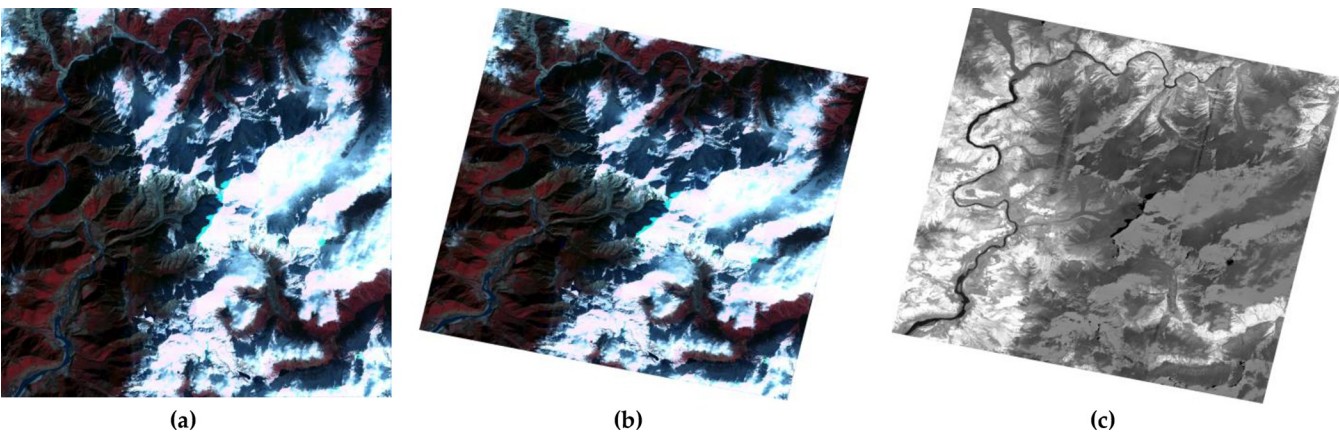

**Fig 5.** GF-1 PMS2 image with center location 95.0˚E, 29.6˚N, recorded on 13nd Dec 2021: (a) Original images; (b) orthorectified fusion image; (c).

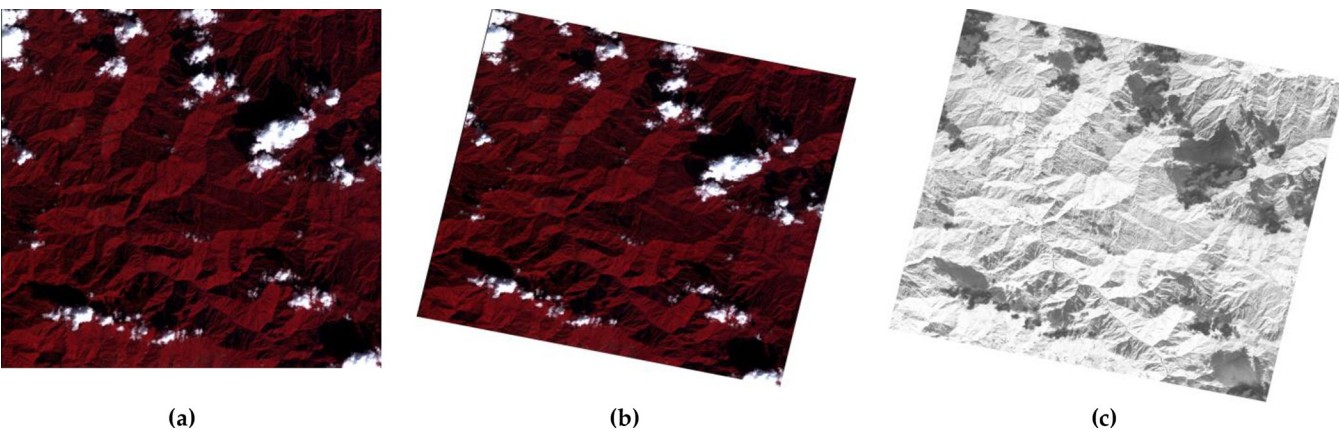

**Fig 6.** GF-2 PMS2 image with center location 93.8˚E, 28.1˚N, recorded on 25th Jan 2019: (a) Original images; (b) orthorectified fusion image; (c) NDVI.

centered around location 93.8˚E, 28.1˚N, on 25th Jan 2019. "A1" is the multispectral image from the PMS and "A2" is the image from the panchromatic band of the PMS. Therefore, we need to do data fusion processing, and then we can get high spatial resolution and high spectral resolution, which is of great help for the image classification.

The results presented in Figs 3–6 are produced using Snowy Dove by executing user-defined commands in a batch file. Snowy Dove produces the same results as manual processing of a series of user-defined commands but more efficient and faster.

## Characteristics

Snowy Dove can run under *nix and Windows thanks to the cross-platform nature of the scripting languages IDL and python. In addition, there are several characteristics of Snowy

**Table 2. Time cost comparison to decompression data of different sizes using different scripts (units: Seconds).**

| data size | 189M | 778M | 1.1G | 3.6G | 4.1G | 10.4G |
|---|---|---|---|---|---|---|
| FILE_UNTER in IDL | 8.84 | 17.81 | 59.87 | 300.10 | 417.94 | 1004.33 |
| tar in *nix (this study) | 8.10 | 15.58 | 22.69 | 144.20 | 190.62 | 457.66 |

**Table 3. Time cost comparison to do orthorectification of different sizes data using different scripts (units: Seconds).**

| data size | MSS | | | PAN | | |
|---|---|---|---|---|---|---|
| | 580M | 1.3G | 4G | 661M | 1.6G | 4.3G |
| ENVI interface in IDL | 155.48 | 502.51 | 4196.47 | 60.66 | 139.66 | 725.51 |
| gdal interface in python3 (this study) | 3.56 | 10.71 | 35.83 | 7.68 | 12.08 | 91.29 |

Dove: it processes data in batch mode, it is developed by multiple languages, it has modular design, and it is open source and is under Long-Term Support (LTS),The algorithms used in Snow Pigeon not only have fast and efficient processing capabilities, but also show excellent compatibility with different satellite data.

## Batch mode

Compared with manual pre-processing of images step by step in an interactive interface (hereinafter collectively referred to as "traditional method"), Snowy Dove has the following advantages: 1) Users can process multiple files continuously without the need of multiple manual calls to a program; 2) Memory-Save. With the traditional method, the graphical user interface (GUI) will take up part of the memory, and running the Snowy Dove program will allocate this part of unnecessary memory to the image pre-processing part and thereby improves efficiency; 3) Fragmentation-Time-Optimization. When using the traditional method, if the image is too large, the time to process each step is more than several hours and users cannot accurately predict the time when the step is completed. Hence, the time between the end of the previous step and the beginning of the next step, will inevitably be fragmented. In addition, if there are many pre-processing steps, the traditional method will generate many temporary files and therefore, when the pre-processing is completed, the storage space needs to be cleaned. Cleaning up will also produce fragmented time. The main fragmentation time of the two parts will slow down the pre-processing process to a large extent. The advantage of Snowy Dove lies in complete elimination of the fragmentation time. Seamless connection between each step can be achieved within one second, and unnecessary temporary files in the intermediate steps will be cleaned up automatically.

## Multiple language

Snowy Dove is developed in multiple languages with the purpose of improving efficiency.

Mixing bash shell script: For users, there are two main ways to decompress the original *.tar.gz data archive. One is to complete the decompression through certain software with GUI, the other is to decompress through commands, which can be decompressed in multiple languages. Table 2 shows the comparison of the cost time of using the FILE_UNTAR function in IDL and the tar command in *nix to decompress some GF archive. Obviously, when the tar command decompresses files exceeding 500 M, its speed is far better than that of the FILE_UNTAR function. Thus, Snowy Dove mixes shell scripts to improve the efficiency of file decompression.

Mixing python script: When performing orthorectification on GF data, experiments have been done with the ENVI Task function in IDL and the GDAL. Warp function in python to perform orthorectification on the experimental data. Table 3 lists the processing time of some data. Using Python + GDAL to perform orthorectification is much faster than ENVI Task. On the other hand, in image fusion there is a fusion algorithm in GDAL. Developers can use the method of exporting files after constructing GDAL Virtual Format (VRT) in python for image fusion. In IDL, developers can also use the ENVI_DOIT function for this proceeding. Fig 7

**Table 4. Time cost comparison of exporting GeoTIFF of different size using different method (units: Seconds).**

| data size | 459M | 1.5G | 7G |
|---|---|---|---|
| ENVIRaster::Export | 5.86 | 18.56 | 462.19 |
| sd_nv2 tiff (this study) | 2.63 | 8.76 | 329.92 |

shows the partial results of image fusion using Python + GDAL and ENVI_DOIT on the experimental data. There is some blurring when using the ENVI interface, as Fig 7 shows, the GDAL algorithm performs better. When GDAL reads the GF data, it can load the RPC information intelligently, while the Openraster function of the ENVI interface can only load the RPC information of the GF-1 and GF-2. To orthorectify the GF-6 image, users need to reload the RPC information manually in advance. If the user wants to use python for orthorectification or image fusion, the current python version needs to be Python3, environmental variables need to be set, and the GDAL library needs to have been installed.

Mixing C-compiled file: C is an efficient language. C program is relatively compact and runs very fast. There are 3 executable files (sd_nv2tiff, sd_radcal and sd_ndvi) supporting Snowy Dove, and each file's source code includes 3 basic headers (stdio.h, stdlib.h and string.h).

sd_nv2tiff: The input parameter of this function is a string file name. Its main function is exporting the ENVI format image to GeoTIFF format image. Under ENVI5.1/IDL8.3, developers can easily call the Export method of the ENVIRaster object to export an image from ENVI format to GeoTIFF, which is very convenient and has good general applicability, but the disadvantage is that it takes a long time. In order to solve the problem that it is time-consuming, a pure IDL method can be used to implement this process using the WRITE_TIFF function, but its drawback is that this function needs to construct the entire image matrix in advance before it can be written to a file, so it is not suitable for computers with low system memory. For these users, it is impossible to perform format conversion by constructing a huge matrix. For example, if an ENVI format image with a 4G storage size is exported as GeoTIFF using the WRITE_TIFF function, a user needs to allocate 4G space in the computer memory

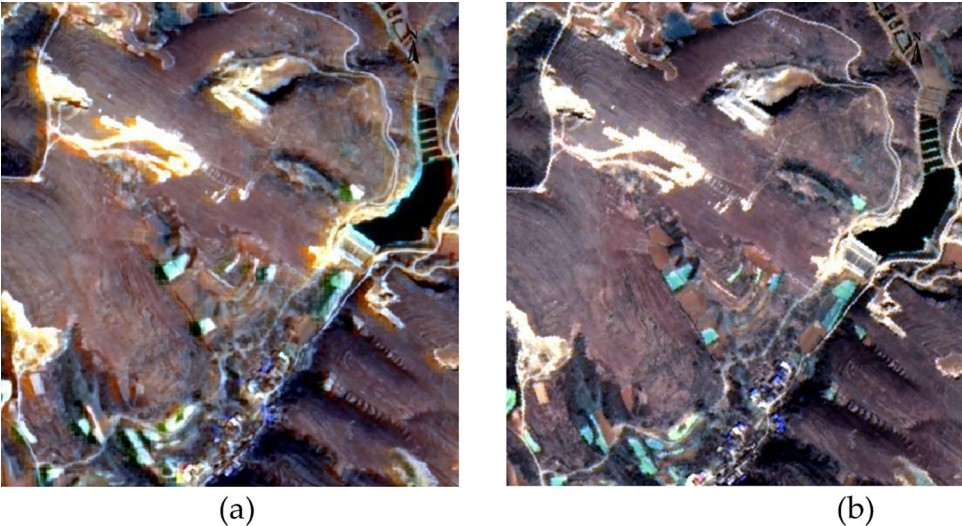

 (a) (b)

**Fig 7.** The performance of different data fusion algorithms on GF-6 PMS data: (a) ENVI algorithm; (b) GDAL algorithm.

**Table 5. Time cost comparison of linear calibration of different size using different method (units: Seconds).**

| data size | MSS | | | PAN | | |
|---|---|---|---|---|---|---|
| | **769M** | **1.5G** | **6G** | **876M** | **1.5G** | **5.9G** |
| ENVI interface in IDL | 46.85 | 98.73 | 944.71 | 53.83 | 110.26 | 660.12 |
| gdal interface in python3 | 16.36 | 39.22 | 424.41 | 17.27 | 35.87 | 487.74 |
| sd_radcal (this study) | 10.32 | 35.51 | 411.91 | 11.81 | 28.68 | 470.33 |

to temporarily store the data, and if the user's system memory has a limit, it cannot be applied and the user cannot perform the exporting operation through incremental writing. In Snowy Dove, the principle of this C-written function is: through the standard I/O file stream, sequentially write Image File Header, image matrix data, and Image File Directory to the output file. Through some local data tests (Table 4), the efficiency of using this method is significantly higher than the ENVIRaster::Export method of the ENVI interface, and it also overcomes the defects of the built-in WRITE_TIFF function in IDL.

sd_radcal: The input parameters of this function are: ① string input file name ② string output file name ③ numeric gain coefficients ④ numeric offset coefficients. Its main function is to achieve the linear scaling processing of ENVI format images. Under ENVI5.1/IDL8.3, developers can easily call the Radiometric Calibration interface from ENVIT ask to perform linear calibration on images, but after testing, this interface does not show a good advantage in terms of efficiency. So, in Snowy Dove, the linear calibration processing function is completed in C to improve efficiency. Table 5 shows the time comparison of the three methods for radiometric calibration of multi-spectral images and panchromatic images of different sizes. Obviously, the ENVI Task interface has the lowest efficiency, while the method of using the bottom I/O for calibration is the most satisfactory. The GDAL function in python is also close to relatively high efficiency.

sd_ndvi: The input parameters of this function are: ① string input file name ② string output file name. Its main function is to calculate the corresponding NDVI image from the ENVI format image. Table 6 shows the time comparison between the ENVI interface and this function for NDVI calculation.

## Modular

The Snowy Dove source code (under /src directory) is composed of multiple functions and multiple files, users can easily locate the places that need to be modified according to their own subjective needs. Moreover, each source file has embedded enough comments, users can easily add more comments, and delete or modify the source code to meet their own needs.

Example 1) If the user needs to process the GF data obtained in 2021, and according to the actual situation, the calibration coefficients in 2021 will not be announced until 2022.

Thus the user has to determine the calibration coefficient by himself. Therefore, in the /src directory, it is easy to change the calibration coefficients in the appropriate file. The center wavelength can also be changed in a similar way.

**Table 6. Time cost comparison of calculating NDVI of different size using different method (units: Seconds).**

| data size | 769M | 1.5G | 6G |
|---|---|---|---|
| ENVI interface in IDL | 3.74 | 7.11 | 13.97 |
| sd_ndvi (this study) | 3.30 | 6.79 | 13.39 |

Example 2) If the user uses ENVI5.3 or a later version, there is a better NNDiffuse algorithm for image fusion and an Application Programming Interface is also provided. So if the user wants to use the NNDiffuse algorithm for image fusion, the only thing needed to do is to open the pansharpen.pro file in the /src directory using a text editor and change the method name.

## Open source & long-term support

Snowy Dove is released on https://github.com/desertstsung/Snowy dove under the MIT license. Anyone who obtains this software has the right to use, copy, modify, merge, publish, distribute, re-authorize and sell the software and a copy of the software. The significance of open source is that users can modify the software as needed to achieve optimal benefits. At the same time, users can also provide comments or corrections to developers according to their actual use, which is conducive to the iterative optimization of the version of the program.

Although this program has passed a lot of test data, it does not rule out the problem of hidden defects. In addition, the calibration coefficients of GF data will change every year. Therefore, Snowy Dove will pay close attention to the user experience and GF calibration coefficients to continuously improve this software.

## Conclusion

In this paper, we present a one-click batch pre-processing toolkit, named Snowy Dove, which we developed for pre-processing of data from the Gaofen series of satellites. Snowy Dove uses multiple programming languages (mainly IDL, and mixing Python, C and Bash Shell). Snowy Dove does not have a user-interactive interface and does not seem to be friendly to users who are not familiar with command-line interface. However, it has strong data processing advantages. First, Snowy Dove can process data more efficiently than some other methods. Second, it can process data in batches and the user has to define what he wants the programme to do, then users can produce multiple files meeting their needs. Third, it is open source and under long term support. In the future, we will continue to develop more functions to achieve compatible processing of more data, making it more complete and easy to use.

## Supporting information

**S1 Appendix.**
(DOCX)

## Author Contributions

**Conceptualization:** Jiantao Dong, Xurui Zhao.

**Data curation:** Liuya Zhang, Jiantao Dong.

**Resources:** Xurui Zhao.

**Supervision:** Debao Yuan.

**Writing – original draft:** Debao Yuan.

**Writing – review & editing:** Liuya Zhang, Jiantao Dong, Cheng Fan, Xurui Zhao.

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
