## [Decision Letter · Decision Letter 0]

15 May 2024

PONE-D-24-13844Snowy Dove: An Open-Source Toolkit for Pre-processing of Chinese Gaofen Series DataPLOS ONE

Dear Dr. Yuan,

Thank you for submitting your manuscript to PLOS ONE. After careful consideration, we feel that it has merit but does not fully meet PLOS ONE’s publication criteria as it currently stands. Therefore, we invite you to submit a revised version of the manuscript that addresses the points raised during the review process.

We look forward to receiving your revised manuscript.

Kind regards,

Bijeesh Kozhikkodan Veettil

Academic Editor

PLOS ONE

Journal Requirements:

   "This work was supported by Hebei Natural Science Foundation Ecological Smart Mine Joint Fund (No. E2020402086) and this work was also supported by National Natural Science Foundation of China (NSFC)(No.52174610)"

6. Please amend the manuscript submission data (via Edit Submission) to include author Jiantao Dong and Xurui Zhao.

Reviewers' comments:

Reviewer's Responses to Questions

**Comments to the Author**

1. Is the manuscript technically sound, and do the data support the conclusions?

Reviewer #1: No

Reviewer #2: Yes

Reviewer #3: No

2. Has the statistical analysis been performed appropriately and rigorously? 

Reviewer #1: No

Reviewer #2: No

Reviewer #3: No

3. Have the authors made all data underlying the findings in their manuscript fully available?

Reviewer #1: Yes

Reviewer #2: No

Reviewer #3: Yes

4. Is the manuscript presented in an intelligible fashion and written in standard English?

Reviewer #1: No

Reviewer #2: Yes

Reviewer #3: Yes

5. Review Comments to the Author

Reviewer #1: Thank you for submitting your manuscript. After thorough review, I have identified several areas that require significant attention before considering your work for publication.

1. Clarity and Writing Quality:

The manuscript suffers from numerous grammatical errors, awkward phrasings, and unclear explanations throughout, making it challenging to follow the authors' reasoning and evaluate the methodology. A comprehensive revision of the language and structure is necessary to meet the standards of scientific publication. Please ensure clarity and coherence in your writing to enhance readability.

2. Novelty and Significance:

The introduction lacks a comprehensive review of existing pre-processing methods for Chinese satellite data, making it unclear how Snowy Dove contributes to advancing the state-of-the-art. Please clearly articulate the specific limitations of current approaches and highlight the unique contributions of your work to establish its novelty and significance.

3. Methodological Details:

The methodology section lacks important technical details on the algorithms implemented in Snowy Dove, particularly concerning the atmospheric correction approach. Please provide a more thorough explanation of your processing algorithms and workflow to enable proper evaluation and reproducibility.

4. Results and Analysis:

The results and analysis presented in the paper are insufficient to demonstrate the effectiveness and advantages of the Snowy Dove toolkit. Please ensure detailed captions and annotations for example images to facilitate interpretation of the improvements achieved by the software. Additionally, provide a more comprehensive and transparent analysis, including quantitative accuracy assessments of the geometric and radiometric corrections.

5. Discussion Depth:

The discussion of the toolkit's characteristics and limitations lacks critical reflection and comparison with other existing software options. Please provide a meaningful analysis of how the capabilities of Snowy Dove compare to those of alternative tools and address potential limitations to enhance the depth of your discussion.

6. Referencing:

The number and quality of references cited in the paper are insufficient. Please conduct a more thorough literature review and engage with relevant studies on satellite data pre-processing workflows, algorithm development, and accuracy assessment to contextualize your work within the broader field and support your methodological choices.

In conclusion, while I recognize the potential value of Snowy Dove as an open-source toolkit for pre-processing Chinese satellite data, the manuscript requires substantial revisions to address the identified issues.

Reviewer #2: i. The manuscript doesn't clarify what makes this toolkit unique. Are there existing pre-processing

methods for Gaofen data? If so, how does this toolkit improve upon them?

ii. To enhance the manuscript's impact, it would be beneficial to include a thorough comparison with existing pre-

processing methods, showcasing the specific advantages and novel features of the proposed toolkit. This would

strengthen the manuscript's originality and highlight its potential impact on the field

iii. The toolkit focus solely on pre-processing steps, potentially excluding image analysis or post-processing

functionalities.

iv. To enhance the manuscript's clarity and usability, it is essential to provide detailed explanations of all preprocessing

algorithms included in the toolkit. This would ensure that users would have have a comprehensive understanding of

the algorithms and can implement them in their own work

Reviewer #3: 1. This paper's references and citations have several concerns. The authors only listed five sources.

2. This paper discusses a number of concepts that either need a definition or a reference be included.

3. Figure quality needs to be at least 300 dpi.

4. What are PAN and MSS? Which kind of statistical parameter is this? Why is it appropriate for this study?

5. As far as pre-processing goes, "APPENDIX A" only requires one click. For this implementation to be shown, the author should add the pseudocodes.

6. What is the experiment on the testbed?

This study requires a thorough correction, in particular the authors should add more citations and compare their findings both qualitatively and quantitatively with those of other comparable studies. Additionally, this study has to contain an experimental testbed in order to replicate this experiment by another researcher.

6. PLOS authors have the option to publish the peer review history of their article (what does this mean?). If published, this will include your full peer review and any attached files.

Reviewer #1: No

Reviewer #2: No

Reviewer #3: No

---

## [Author Response · Author response to Decision Letter 0]

26 Aug 2024

Reviewer #1: 

Thank you for submitting your manuscript. After thorough review, I have identified several areas that require significant attention before considering your work for publication.

Thank you very much for your careful review, we will definitely revise it carefully and strive to publish it as soon as possible.

1. Clarity and Writing Quality:

The manuscript suffers from numerous grammatical errors, awkward phrasings, and unclear explanations throughout, making it challenging to follow the authors' reasoning and evaluate the methodology. A comprehensive revision of the language and structure is necessary to meet the standards of scientific publication. Please ensure clarity and coherence in your writing to enhance readability.

We have made a complete revision of the language and structure of the manuscript, as well as the logical structure of the manuscript.

2. Novelty and Significance:

The introduction lacks a comprehensive review of existing pre-processing methods for Chinese satellite data, making it unclear how Snowy Dove contributes to advancing the state-of-the-art. Please clearly articulate the specific limitations of current approaches and highlight the unique contributions of your work to establish its novelty and significance.

In the introduction, we introduce a series of current preprocessing methods for domestic satellites, including the processing of each scene image by ENVI software, the extraction of grape growing areas by #C language, and the data processing of individual satellites by IDL language and so on. At present, there is no tool that can realize the batch processing of all the pretreatment processes of GF1/2/6, etc., which is the great significance of the white pigeon proposed in this study.

3. Methodological Details:

The methodology section lacks important technical details on the algorithms implemented in Snowy Dove, particularly concerning the atmospheric correction approach. Please provide a more thorough explanation of your processing algorithms and workflow to enable proper evaluation and reproducibility.

Thanks for the suggestion, we've added more technical details in the methods section, especially for atmospheric correction.

4. Results and Analysis:

The results and analysis presented in the paper are insufficient to demonstrate the effectiveness and advantages of the Snowy Dove toolkit. Please ensure detailed captions and annotations for example images to facilitate interpretation of the improvements achieved by the software. Additionally, provide a more comprehensive and transparent analysis, including quantitative accuracy assessments of the geometric and radiometric corrections.

Thanks to the suggestion, we have redrawn the picture of the article, and also supplemented it with a detailed introduction and quantitative analysis.

5. Discussion Depth:

The discussion of the toolkit's characteristics and limitations lacks critical reflection and comparison with other existing software options. Please provide a meaningful analysis of how the capabilities of Snowy Dove compare to those of alternative tools and address potential limitations to enhance the depth of your discussion.

In Tables 2 and 3, we have made a comparative analysis of the processing efficiency of different file sizes, and give a quantitative evaluation, which can highlight the important value of the Snowy Dove.

6. Referencing:

The number and quality of references cited in the paper are insufficient. Please conduct a more thorough literature review and engage with relevant studies on satellite data pre-processing workflows, algorithm development, and accuracy assessment to contextualize your work within the broader field and support your methodological choices.

Thanks for the comments, we have re-supplemented the exhaustive references, and we hope you are satisfied.

In conclusion, while I recognize the potential value of Snowy Dove as an open-source toolkit for pre-processing Chinese satellite data, the manuscript requires substantial revisions to address the identified issues.

Thank you for your recognition of Snowy Dove, according to the reviewers' comments, we have made comprehensive revisions and improvements to the manuscript.

Reviewer #2: 

i. The manuscript doesn't clarify what makes this toolkit unique. Are there existing pre-processing methods for Gaofen data? If so, how does this toolkit improve upon them?

1.There are existing tools for the processing steps, like GUI tool ENVI. But the pro-processing steps are split into individual tools, users must use multiple tools to accomplish the pro-processing pipeline, and as each file size of a certain scene of Gaofen data is enormous, the processing time always cost hours in a GUI way. This toolkit will reduce the time between individual steps into nearly zero, and could processing as many data as user inputs.

2.The main improvement is that we optimize some algorithms from scratch, which is faster than the corresponding function in native ENVI/IDL. As for other processing steps, the toolkit is smart enough to choose a different way in order to save time, while ensuring the results are correct. For example, if the host OS is based on *nix, the de-compression step would use the `tar` toolkits instead of native IDL function.

ii. To enhance the manuscript's impact, it would be beneficial to include a thorough comparison with existing pre-processing methods, showcasing the specific advantages and novel features of the proposed toolkit. This would strengthen the manuscript's originality and highlight its potential impact on the field

iii. The toolkit focus solely on pre-processing steps, potentially excluding image analysis or post-processing functionalities.

Thanks for the suggestion. This toolkit do focus on pre-processing steps, which are less subjective than post-processing, and the post-processing steps differs across a range of application areas. So, we develop this toolkit first, and we will consider the later one.

iv. To enhance the manuscript's clarity and usability, it is essential to provide detailed explanations of all preprocessing algorithms included in the toolkit. This would ensure that users would have have a comprehensive understanding of the algorithms and can implement them in their own work.

Thanks for the suggestion. We provided some detailed explanations of all preprocessing algorithms included in the toolkit.

Reviewer #3:

1. This paper's references and citations have several concerns. The authors only listed five sources.

Thanks for the suggestions, we have update the references and add some latest citation.

2. This paper discusses a number of concepts that either need a definition or a reference be included.

Yes, we added both definition and reference for the concepts in the paper.

3. Figure quality needs to be at least 300 dpi.

Thanks for the suggestion. We have updated the figure in the manuscript.

4. What are PAN and MSS? Which kind of statistical parameter is this? Why is it appropriate for this study?

In the data part, we wrote “PAN and MSS represent panchromatic bands and multi-spectral band”， PAN and MSS are the two camera onboarded the GF1, GF2 and GF6. However the resolution of PAN and MSS of GF-2 is 1 m and 4 m, which is different from the 2 m and 8 m of GF-1 and GF-6. The data acquired by these two cameras was the main source of data for this study.

5. As far as pre-processing goes, "APPENDIX A" only requires one click. For this implementation to be shown, the author should add the pseudocodes.

Thanks for the suggestion, we added the pseudocodes as APPENDIX B, hope you are satisfied with the revised manuscript.

6. What is the experiment on the testbed? This study requires a thorough correction, in particular the authors should add more citations and compare their findings both qualitatively and quantitatively with those of other comparable studies. Additionally, this study has to contain an experimental testbed in order to replicate this experiment by another researcher.

We have tested this toolkit under Windows and Linux-based x86 operating systems, like Ubuntu and Fedora, etc. as for software environment, the basic one is the version of ENVI/IDL must be greater than 5.3/8.1, we tested different ENVI version, and if Python/GDAL installed. However, thanks for the cross-platform ability of ENVI/IDL and Python, this toolkit should run perfectly on macOS, too, we welcome Apple users pull their contributions. As we mentioned, ENVI/IDL, which is a commercial software package, plays an essential role in this toolkit, therefore, we could only provide source codes written in IDL with `.pro` filename suffix, which are also publicly available.

---

## [Decision Letter · Decision Letter 1]

15 Sep 2024

PONE-D-24-13844R1Snowy Dove: An Open-Source Toolkit for Pre-processing of Chinese Gaofen Series DataPLOS ONE

Dear Dr. Yuan,

Thank you for submitting your manuscript to PLOS ONE. After careful consideration, we feel that it has merit but does not fully meet PLOS ONE’s publication criteria as it currently stands. Therefore, we invite you to submit a revised version of the manuscript that addresses the points raised during the review process.

We look forward to receiving your revised manuscript.

Kind regards,

Bijeesh Kozhikkodan Veettil

Academic Editor

PLOS ONE

Journal Requirements:

Reviewers' comments:

Reviewer's Responses to Questions

**Comments to the Author**

1. If the authors have adequately addressed your comments raised in a previous round of review and you feel that this manuscript is now acceptable for publication, you may indicate that here to bypass the “Comments to the Author” section, enter your conflict of interest statement in the “Confidential to Editor” section, and submit your "Accept" recommendation.

Reviewer #2: All comments have been addressed

Reviewer #3: All comments have been addressed

2. Is the manuscript technically sound, and do the data support the conclusions?

Reviewer #2: No

Reviewer #3: Yes

3. Has the statistical analysis been performed appropriately and rigorously? 

Reviewer #2: No

Reviewer #3: Yes

4. Have the authors made all data underlying the findings in their manuscript fully available?

Reviewer #2: No

Reviewer #3: Yes

5. Is the manuscript presented in an intelligible fashion and written in standard English?

Reviewer #2: No

Reviewer #3: (No Response)

6. Review Comments to the Author

Reviewer #2: 1. The research project lacks explicit methods and algorithms for L0 and L1 level products

2. There are several formatting inconsistencies that need to be addressed to align with the journal's guidelines.

I recommend that the authors take the time to address these points before resubmitting the manuscript for further consideration

Reviewer #3: (No Response)

7. PLOS authors have the option to publish the peer review history of their article (what does this mean?). If published, this will include your full peer review and any attached files.

Reviewer #2: No

Reviewer #3: No

---

## [Author Response · Author response to Decision Letter 1]

3 Oct 2024

Reviewer #2: 

1.The research project lacks explicit methods and algorithms for L0 and L1 level products

Many thanks to the reviewers for their valuable comments. Regarding the explicit methodology and algorithms for the L0 and L1 level products, I would like to clarify that in our study, the L0 and L1 level data were pre-prepared and provided by external sources. The focus of this study is to utilize these pre-existing L1 level products as input to the algorithm for subsequent analysis and processing. Therefore, the L0 to L1 conversion process is not within the direct scope of this study. Our work mainly focuses on how to utilize these L1-level data to achieve the research objectives through our proposed specific algorithm. Once again, we thank the reviewers for their review and suggestions, and we will continue to work on improving the study.

2. There are several formatting inconsistencies that need to be addressed to align with the journal's guidelines.

Thank you very much to the reviewers for their meticulous review and for pointing out the formatting inconsistencies in our manuscript. We understand that formatting consistency is critical to ensuring a professional and readable article and fully appreciate the importance of following journal guidelines.

In response to the issues raised by the reviewers, we will immediately take the following steps to rectify the situation:

Detailed review of the journal guidelines: We will read and understand the journal's formatting requirements again to ensure that we have a clear understanding of each stipulation.

Comprehensively check and correct formatting: In accordance with the journal's guidelines, we will conduct a comprehensive formatting check of the manuscript, including but not limited to fonts, font sizes, line spacing, paragraph formatting, header hierarchy, citation formatting, and presentation of charts, graphs and tables, etc. We will also check and correct the formatting of the manuscript. We will check and correct all non-conformities on a case-by-case basis.

Use of professional tools to assist: In order to ensure consistency and accuracy of formatting, we will utilize professional typesetting tools (e.g. LaTeX, Word's style function, etc.) to assist in the revision process and make final adjustments with the help of templates provided by the journal .

Re-review and proofreading: After the revisions are completed, we will conduct several rounds of review and proofreading to ensure that all formatting issues have been properly addressed and that the entire manuscript is formatted to meet the journal's publication standards.

We understand the importance of formatting corrections and appreciate the reviewers' valuable input in this regard. We are committed to finalizing these revisions as soon as possible and submitting the revised manuscript for further processing as soon as possible.

Once again, we thank the reviewers for their review and valuable suggestions, and we look forward to continuing to receive your guidance and support.

I recommend that the authors take the time to address these points before resubmitting the manuscript for further consideration

---

## [Decision Letter · Decision Letter 2]

15 Oct 2024

PONE-D-24-13844R2Snowy Dove: An Open-Source Toolkit for Pre-processing of Chinese Gaofen Series DataPLOS ONE

Dear Dr. Yuan,

Thank you for submitting your manuscript to PLOS ONE. After careful consideration, we feel that it has merit but does not fully meet PLOS ONE’s publication criteria as it currently stands. Therefore, we invite you to submit a revised version of the manuscript that addresses the points raised during the review process.

We look forward to receiving your revised manuscript.

Kind regards,

Bijeesh Kozhikkodan Veettil

Academic Editor

PLOS ONE

Journal Requirements:

Reviewers' comments:

Reviewer's Responses to Questions

**Comments to the Author**

1. If the authors have adequately addressed your comments raised in a previous round of review and you feel that this manuscript is now acceptable for publication, you may indicate that here to bypass the “Comments to the Author” section, enter your conflict of interest statement in the “Confidential to Editor” section, and submit your "Accept" recommendation.

Reviewer #2: (No Response)

2. Is the manuscript technically sound, and do the data support the conclusions?

Reviewer #2: Partly

3. Has the statistical analysis been performed appropriately and rigorously? 

Reviewer #2: No

4. Have the authors made all data underlying the findings in their manuscript fully available?

Reviewer #2: No

5. Is the manuscript presented in an intelligible fashion and written in standard English?

Reviewer #2: Yes

6. Review Comments to the Author

Reviewer #2: I recommend to integrate your previous responses into the revised manuscript (i.e. the novelty of your research and detail explanation of the algorithms and why you have chosen them.

7. PLOS authors have the option to publish the peer review history of their article (what does this mean?). If published, this will include your full peer review and any attached files.

Reviewer #2: No

---

## [Author Response · Author response to Decision Letter 2]

21 Oct 2024

Reviewer #2: I recommend to integrate your previous responses into the revised manuscript (i.e. the novelty of your research and detail explanation of the algorithms and why you have chosen them.

Dear Reviewer #2,

Thank you for your valuable feedback and recommendation. I appreciate your suggestion to integrate my previous responses into the revised manuscript.

In response to your comment, I have revised the manuscript to include a more detailed explanation of the novelty of our research and the algorithms used in our toolkit. Specifically, I have highlighted how our toolkit integrates and optimizes Gaofen data preprocessing steps, reducing processing time and enabling batch processing for GF1/2/6 data, which is a significant advancement over existing tools like ENVI. Additionally, I have provided a more detailed explanation of the algorithms we have optimized from scratch and why we chose them, emphasizing their superiority in terms of speed and accuracy compared to the corresponding functions in native ENVI/IDL.

For raising the concern regarding the availability of all data found in our manuscript.

We understand your concern and want to assure you that we have taken steps to ensure that all relevant data and code are accessible to readers. As you mentioned, we have included the relevant code in the appendix of the manuscript. Additionally, we have provided a link to the data in the manuscript,(The GaoFen imagery used in this research is supported by the China Centre for Resources Satellite Data and Application (http://www.cresda.com, accessed on 5 July 2024).) which we believe provides easy access for interested readers to obtain the data for further analysis or verification.

We believe that providing the code and data in this manner adheres to best practices in research transparency and reproducibility. However, if there are specific data points or analyses that you believe are missing or not fully described, please let us know, and we will be happy to provide additional information or clarification as needed.

I believe these revisions will help to clarify the contributions of our research and strengthen the manuscript. Thank you again for your constructive feedback, and I look forward to your further comments on the revised manuscript.

---

## [Editor Report · Decision Letter 3]

28 Oct 2024

Snowy Dove: An Open-Source Toolkit for Pre-processing of Chinese Gaofen Series Data

PONE-D-24-13844R3

Dear Dr. Yuan,

We’re pleased to inform you that your manuscript has been judged scientifically suitable for publication and will be formally accepted for publication once it meets all outstanding technical requirements.

Kind regards,

Bijeesh Kozhikkodan Veettil

Academic Editor

PLOS ONE
---

## [Editor Report · Acceptance letter]

4 Nov 2024

PONE-D-24-13844R3 

PLOS ONE

Dear Dr. Yuan, 

I'm pleased to inform you that your manuscript has been deemed suitable for publication in PLOS ONE. Congratulations! Your manuscript is now being handed over to our production team.

Kind regards, 

on behalf of

Dr. Bijeesh Kozhikkodan Veettil 

Academic Editor

PLOS ONE